# *SfDicer2* RNA Interference Inhibits Molting and Wing Expansion in *Sogatella furcifera*

**DOI:** 10.3390/insects13080677

**Published:** 2022-07-27

**Authors:** Qing-Hui Zeng, Gui-Yun Long, Xi-Bin Yang, Ze-Yan Jia, Dao-Chao Jin, Hong Yang

**Affiliations:** Guizhou Provincial Key Laboratory for Agricultural Pest Management of the Mountainous Region, Scientific Observing and Experimental Station of Crop Pests in Guiyang, Ministry of Agriculture and Rural Affairs, Institute of Entomology, Guizhou University, Guiyang 550025, China; 18311516405@163.com (Q.-H.Z.); lgy0256@126.com (G.-Y.L.); yangxibingz@126.com (X.-B.Y.); jiazeyan520@163.com (Z.-Y.J.); daochaojin@126.com (D.-C.J.)

**Keywords:** *Sogatella furcifera*, *Dicer2*, RNA interference, molting, wing expansion

## Abstract

**Simple Summary:**

Endoribonuclease 2 (Dicer2) plays various physiological roles in the RNA interference (RNAi) pathway by fragmenting double-stranded RNA to generate small interfering RNA, which then mediates gene silencing. In this study, the role of *Dicer2* in the regulation of molting and wing expansion in *Sogatella furcifera* (white-backed planthopper) was investigated. In particular, *SfDicer2*-mediated RNAi resulted in wing deformities and lethal modifications in *S. furcifera*, which are attributable to the significant inhibition of chitin synthesis and degradation and wing expansion genes. This study provides insights into the biological functions of *Dicer2* in insects, which can aid in RNAi-mediated pest control.

**Abstract:**

Endoribonuclease 2 (Dicer2) is a key nicking endonuclease involved in the small interfering RNA biosynthesis, and it plays important roles in gene regulation and antiviral immunity. The Dicer2 sequence was obtained using the transcriptomic and genomic information of *Sogatella furcifera* (Horváth), and the spatiotemporal characteristics and functions of molting and wing expansion regulation were studied using real-time quantitative polymerase chain reaction and RNA interference (RNAi) technology. The expression of *SfDicer2* fluctuated during the nymphal stage of *S. furcifera*. Its expression decreased significantly over the course of molting. *SfDicer2* exhibited the highest transcript level in the nymphal stage and adult fat body. After *SfDicer2* was silenced, the total mortality rate was 42.69%; 18.32% of the insects died because of their inability to molt. Compared with the effects of ds*GFP* or water, 44.38% of the insects subjected to the silencing of *SfDicer2* exhibited wing deformities after successful eclosion. After *SfDicer2* RNAi, the expression of chitinase, chitin deacetylase, trehalase, chitin synthase 1, and wing expansion-related genes was significantly inhibited. These findings indicate that *SfDicer2* controls molting by affecting genes associated with chitin synthesis and degradation and regulates wing expansion by altering the expression of wing expansion-related genes in *S. furcifera*.

## 1. Introduction

RNA interference (RNAi) can reduce the transcript levels of target genes and facilitate the screening of candidate genes for pest control [1,2]. Based on the mechanism of small RNA synthesis, RNAi pathways can be one of three types: small interfering RNA (siRNA), non-coding microRNA (miRNA), and PIWI-interacting RNA pathways [3,4,5,6]. Studies on insects typically employ the siRNA pathway for RNAi. After the accumulation of double-stranded RNAs (dsRNA) in insect cells, they are largely fragmented into siRNAs using endoribonuclease 2 (Dicer2) and are merged with Argonaute 2 (AGO2) to form the RNA-induced silencing complex (RISC), which is responsible for the silencing of target genes at the post-transcriptional level and the regulation of gene expression [7,8,9]. siRNAs are processed from dsRNAs encoded by viruses or introduced exogenously, whereas miRNAs are generally transcribed from endogenous genes [10]. Therefore, most studies on the biological function of small RNAs have focused on miRNAs. However, some studies have demonstrated that the endogenous siRNA-mediated RNAi pathway can regulate the activity of transposons and the expression of protein-coding genes [11].

The Dicer2 protein is an essential component of RISC, which indicates that the protein plays a central role in the initiation stage of RNAi and participates in the effect stage. A study on fruit flies (*Drosophila melanogaster*) revealed that Dicer2 interacted with the dsRNA-binding protein R2D2 to form RISC, identified the kinetic stability of the two ends of siRNA, and selected one of the chains as the RISC-guiding chain [12]. Although Dicer2 can fragment dsRNA into siRNA, siRNA must form a Dicer2–R2D2 complex with R2D2 (the dsRNA-binding domain of R2D2 binds to siRNA and promotes the assembly of siRISC) to mediate RNAi before it can be delivered into the RNAi pathway. The evidence provided by Kalidas et al. supports this view [13]. Adding dsRNA to the extracts of *R2D2* mutants leads to the production of siRNA; however, the siRNA cannot be assembled onto siRISC. At this time, if the *R2D2* mutant is transferred again, the ability of siRNA to bind to siRISC returns to normal. When dsRNA is added to the extracts of *Dicer2* mutants, no siRNA is produced; *R2D2* mutants are undetectable in the *Dicer2* mutant extracts. Thus, *Dicer2* can fragment dsRNA into siRNA and affect the stability of *R2D2*, and *R2D2* can guide the assembly of siRNA into siRISC. Therefore, both *R2D2* and *Dicer2* are required for the assembly of siRNAs into siRISC [12,13]. Although the *Dicer2* homozygous mutant of *D**. melanogaster* could survive and reproduce [14,15], a gene expression profiling of the pupae of *D**. melanogaster Dicer2* deletion mutants uncovered 306 upregulated and 357 downregulated genes, most of which were associated with energy metabolism as well as growth and development. In addition, the pupae of *Dicer2* deletion mutants had lower ATP levels than those of control insects, indicating that the *Dicer2* deletion mutant of *D**. melanogaster* had impaired mitochondrial productivity [16]. However, silencing *Dicer2* in the brown rice planthopper (*Nilaparvata lugens* [Stål]) did not affect its growth and development [17]. To date, few studies have examined the specific effects of *Dicer2* during the growth and development of other insects.

*Sogatella furcifera*, a member of the order Hemiptera and family Delphacidae, is a major migratory pest of rice that causes significant economic losses when its infestation is severe [18]. As a paurometabolous insect, *S. furcifera* grows and develops in three stages: egg, nymph, and adult. *S. furcifera* adults have two types of wings, namely, long and short, with strong adaptability [19]. Research on the genes regulating molting and wing expansion in *S. furcifera* will help identify and develop new potential target genes for controlling and preventing *S. furcifera* infestation. Dicer2 is an essential element that serves as the “first knife” in the RNAi pathway. It is mainly involved in the synthesis of siRNA, which causes the specific degradation of homologous mRNA. Based on published transcriptome [20] and genome [21] data, *SfDicer2* was explored to determine its role in the molting and wing expansion of *S. furcifera* using an RNAi method. The results may help devise novel strategies for pest control based on RNAi.

## 2. Materials and Methods

### 2.1. Laboratory Insects

An *S. furcifera* cohort was collected from a rice field in Huaxi, Guiyang, Guizhou, China (26°31.302″ N and 106°62.294″ E), in 2013 and reared on Taichung Native 1 rice plants at the tillering stage without exposure to any pesticide in the laboratory. Fresh rice seedlings were replaced every 2 days. The environmental conditions were as follows: temperature, 25 °C ± 1 °C; relative humidity, 70% ± 5%; and photoperiod, 16 h light/8 h dark. The insects were kept in an isolated area so that 1-day-old first-instar nymphs could be used as the source of test insects.

### 2.2. Collection of Samples from Various Developmental Stages and Tissues

We prepared *S. furcifera* samples to assess the expression of *SfDicer2* in different developmental stages and tissues. Samples (n = 15–50) were collected at 28 different time points as follows: the first instar at days 1 and 2 (n = 50); second instar at days 1 and 2 (n = 50); third instar at days 1, 2, and 3 (n = 40); fourth instar at days 1, 2, and 3 (n = 35); fifth instar at days 1, 2, and 3 (n = 25); before molting (n = 20); during molting (n = 20); after molting (n = 20); and male and female adults at 12, 24, 36, 48, 72, and 96 h after eclosion (n = 15). The different tissue samples used in this study included the head (n = 50), integument (n = 30), fat body (n = 100), gut (n = 100), legs (n = 30), wings (n = 30), testes (n = 50), and ovaries (n = 50) from composite samples of 1- to 3-day-old fifth-instar nymphs and 1- to 4-day-old adults. All samples were stored in phosphate-buffered saline (PBS; pH = 7.4). All sampling procedures were repeated thrice to ensure the accuracy of the data. The samples were stored at −80 °C.

### 2.3. Total RNA Extraction and cDNA Synthesis

The total RNA was extracted from the collected samples using the HP Total RNA Kit (Omega Bio-Tek, Norcross, GA, USA) according to the manufacturer’s instructions. The concentration and purity of 1 μL of an RNA sample were tested using the NanoDrop 2000 spectrophotometer (Thermo Fisher Scientific, Waltham, MA, USA), and the integrity was tested using 1.2% agarose gel. Better quality RNA is generally indicated by an OD260/OD280 ratio of 1.8–2.0 and prominent rRNA bands (28S and 18S rRNA) under ultraviolet light after electrophoresis. The RNA samples that met the suitable quality standards were synthesized into first-strand cDNA using the PrimeScript RT Reagent Kit with gDNA Eraser (Takara, Dalian, China) according to the manufacturer’s instructions and stored at −20 °C for subsequent use.

### 2.4. Sequencing Analysis, Design, and Synthesis of Specific Primers

Based on the transcriptome data (GenBank accession number: SRP116252) [20], a portion of the *SfDicer2* fragment was used as a template to search the entire genome (GenBank accession number: PRJNA331022) [21] of *S. furcifera*. We used the DNAMAN 7.0 software (Lynnon Biosoft, San Ramon, CA, USA) to deduce the amino acid sequence. After the Dicer2 protein encoded by *SfDicer2* was verified and analyzed using the BLAST tool on the National Center for Biotechnology Information (NCBI) website (https://blast.ncbi.nlm.nih.gov/Blast.cgi, accessed on 5 June 2022), the ORF Finder tool (https://www.ncbi.nlm.nih.gov/orffinder/ accessed on 5 June 2022) on the NCBI website was employed to identify the open reading frames (ORFs) of *SfDicer2*. ProtParam (https://web.Expasy.org/protparam/ accessed on 6 June 2022) was used to predict the molecular composition, relative molecular mass, isoelectric point, and other physicochemical properties of the amino acids of encoded proteins. Further, the ScanProsite database (https://prosite.expasy.org/ accessed on 6 June 2022) was used to predict conserved domains. We constructed a phylogenetic tree with 1000 bootstrap replications via the neighbor-joining method using the MEGA 6.06 software [22].

The complete sequence of *SfDicer2* was obtained. Primer Premier 6.0 was used to design the specific primers (Table 1). A BLAST analysis against a genomic assembly of *S. furcifera* showed that there was no other sequence in the assembly that was more than 18 bp completely identical between the target gene dsRNA and the genome assembly to avoid off-target effects. The specific primers were synthesized by Tsingke Biotechnology Co., Ltd. (Beijing, China).

### 2.5. Spatiotemporal Representation Analysis of SfDicer2

According to the specific primers designed for a real-time quantitative polymerase chain reaction (RT-qPCR) of the target gene (Table 1), the spatiotemporal expression of *SfDicer2* was examined using ribosomal protein L9 (*Sf**RPL9*, GenBank accession number: KM885285) and alpha 1-tubulin (*Sf**TUB*, GenBank accession: KP735521) [23] as the internal reference genes. Using the synthesized cDNA as a template, three biological and technical repeats were generated for each sample. RT-qPCR was performed using the CFX96TM Real-time Quantitative PCR System (Bio-Rad, Hercules, CA, USA) and 20 μL of the reaction system, comprising 10 μL of FastStart Essential DNA Green Master Mix, 1 μL each of the upstream and downstream primers (10 µM), 1 μL of the *S. furcifera* cDNA template, and 7 μL of ddH_2_O. The following PCR-amplification program was used: pre-denaturation at 95 °C for 10 min; 40 cycles of denaturation at 95 °C for 30 s and annealing and extension at 60 °C for 30 s; and dissolution curve analysis at 60–95 °C.

### 2.6. dsRNA Synthesis

Based on the obtained sequence of *SfDicer2*, gene-specific primers containing the T7 polymerase promoter sequence were designed using the Primer Premier 6.0 software. The dsRNA primers (Table 1) were designed and synthesized and were verified using PCR amplification. The total reaction volume (25 μL) included 1 μL each of the upstream and downstream primers (10 µM), 12.5 μL of the PCR Mix, 3 μL of the *S. furcifera* cDNA template, and 7.5 μL of ddH_2_O. The PCR-amplification conditions were as follows: pre-denaturation at 95 °C for 3 min; 30 cycles of denaturation at 95 °C for 30 s, annealing at 55 °C for 30 s, and extension at 72 °C for 1–3 min; extension at 72 °C for 10 min; and storage at 4 °C. After the PCR products were separated using 1.2% agarose gel electrophoresis, they were subjected to purification and recycling using the E.Z.N.A. Gel Extraction Kit (Omega Bio-Tek). The recovered product was subcloned into the pMD18-T vector (Takara, Dalian, China) and transformed into *Escherichia coli* DH5αcompetent cells (TransGen Biotech, Beijing, China), and the colonies were selected and sent to Tsingke Biotechnology Co., Ltd. for sequencing. The correct sequence was amplified in the cultivated liquid medium, and the plasmid was extracted using the Plasmid Midi Kit (Omega Bio-Tek). The dsRNA was synthesized and purified in vitro using the TranscriptAid T7 High Yield Transcription Kit and GeneJET RNA Purification Kit (both from Thermo Fisher Scientific), according to the manufacturer’s instructions, using the plasmid as the cDNA template for PCR and the recovered high-concentration dsDNA as the template. After synthesis, the dsRNA was diluted, and the concentration was determined by the extinction coefficient RNA using a NanoDrop 2000 spectrophotometer. Subsequently, the integrity of dsRNA was determined using 1.2% agarose gel electrophoresis. The green fluorescent protein dsRNA (ds*GFP*) (GenBank accession number: CAA58789) was then synthesized using the same method, stored at −80 °C, and used as a non-specific negative control.

### 2.7. SfDicer2 RNAi Analysis

Same-sized healthy nymphs were selected to be injected with the synthesized dsRNA. The *S. furcifera* specimens were anesthetized for 40 s using CO_2_, after which the test insects were placed on a 2% agarose plate with the abdomen facing upward. The dsRNA was injected using an IM-31 microinjector (Narishige, Tokyo, Japan) at a point where the anterior and middle chests were connected. A gentle injection was required to avoid mechanical damage to the test insects. In total, 0.1 μL (approximately 100 ng) of dsRNA was injected into the 1-day-old fourth-instar specimens. The nuclease-free water and ds*GFP* groups were administered to the insects in the negative control groups. Each group comprised 120 nymphs, and all treatments were performed in triplicates. After injection, the test insects were transferred to a test tube containing fresh rice seedlings and placed in an artificial climate box.

Samples were collected every 24 h after injection for 3 consecutive days. Total RNA was extracted from the 10 surviving insects in each group and was reverse-transcribed into cDNA. RT-qPCR was used to detect the gene silencing’s efficiency after *SfDicer2* interference. The phenotypes of *S. furcifera* in each group were observed every day, and mortality was examined for 8 consecutive days. All samples were collected and soaked in PBS, and the phenotype of *S. furcifera* was observed, compared, and photographed under a stereoscopic microscope (SMZ25 Nikon Corporation, Tokyo, Japan).

### 2.8. Analysis of the Effects of the Silencing of SfDicer2 on Chitin and Wing Expansion-Related Genes

To verify the effect of *SfDicer2* on chitin (chitinase, chitin deacetylase, trehalase, and chitin synthase 1) and wing expansion in *S. furcifera*, the mRNA expression of these pathways and genes was detected using RT-qPCR. *Sf**RPL9* and *Sf**TUB* were used as the internal references. Information regarding the fluorescence quantitative primers is presented in Table 1.

### 2.9. Statistical Analyses

Using the 2^−∆∆CT^ method [24], we evaluated the transcript levels of *SfDicer2*, along with chitinase, chitin deacetylase, trehalase, and chitin synthase 1, and the wing expansion-related genes. All data were analyzed using SPSS 22.0 (SPSS, Chicago, IL, USA). The values are presented as the mean ± standard error (SE) of three replicates. One-way analysis of variance and Duncan’s test were used to compare the relative expression of each sample in various developmental stages, molting stages, tissues, and the three experimental conditions (*p* < 0.05). After silencing *SfDicer2*, the expression of the target genes, including chitinase, chitin deacetylase, trehalase, chitin synthase, and wing expansion-related and other related genes was evaluated and compared with that of the genes in the control(ds*GFP*) group. A Student’s *t*-test for independent samples was performed to assess statistical significance, which was indicated by *p* < 0.05.

## 3. Results

### 3.1. Sequence Characterization and Phylogenetic Analysis of SfDicer2

A 6818-bp gene sequence was obtained from the genomic and transcriptomic data of *S. furcifera*. An NCBI alignment identified the gene as *SfDicer2* (GenBank accession number: ON693521). The *SfDicer2* cDNA sequence contained a complete 6558-bp ORF encoding a hypothetical protein sequence of 2185 amino acids (Figure 1A). The results predicted by the online software ProtParam revealed that the molecular weight of *SfDicer2* was approximately 250.15 kDa and that the theoretical isoelectric point was 7.59. The instability index was computed to be 43.68, which marked the protein as unstable.

We investigated the degree of similarity between the Dicer2 protein of *S. furcifera* and that of other species using a BLAST homology search and comparison analysis. The Dicer2 protein of *S. furcifera* showed high similarities of 81.49% and 75.81% with that of *Laodelphax striatellus* (GenBank accession number: AGE12616.1) and *N. lugens* (GenBank accession number: AFK73581.1), respectively, although *S. furcifera* is distantly related to other insects. The phylogenetic tree revealed that the Dicer2 protein of *S. furcifera* was most closely related to that of *N. lugens* and *L. striatellus* (Figure 1B). A domain analysis revealed that the Dicer2 protein had seven typical conservative domains in three rice planthoppers: a superfamilies one and two helicase ATP-binding type-1 domain profile (HELICASE_ATP_BIND_1), a superfamilies one and two helicase C-terminal domain profile (HELICASE_CTER), a Dicer double-stranded RNA-binding fold domain profile (DICER_DSRBF), a PAZ domain profile (PAZ), two ribonuclease III family domain profiles (RNase III), and a double-stranded RNA-binding domain profile (dsRBD). In addition, seven conserved domains that were key to the catalytic activity of Dicer2 enzymes were identified (Figure 1C).

### 3.2. SfDicer2 Expression in Various Developmental Stages and Tissues

*SfDicer2* was expressed in all developmental stages of *S. furcifera*. In particular, the expression of *SfDicer2* was high in 1-day-old first-instar nymphs, 2- and 3-day-old third-instar nymphs, and 1- and 3-day-old fifth-instar nymphs, indicating that *SfDicer2*’s expression fluctuated in the nymphal stage (Figure 2A, Appendix A). Based on the findings before, during, and after molting, the expression of *SfDicer2* decreased with increasing molting time from the nymphal stage to the adult stage (*p* < 0.05; Figure 2B, Appendix A). *SfDicer2*’s expression in different tissues of the fifth-instar nymphs decreased in the order of fat body = head > gut = integument > legs (Figure 2C, Appendix A). Moreover, its expression in the fat body of adults was significantly higher than that of the head, wing, testis, gut, and ovary, but there was no significant difference compared with the integument (*p* > 0.05; Figure 2D, Appendix A).

### 3.3. SfDicer2 RNAi Causes Wing Expansion Failure and Death in S. furcifera

To explore whether the decrease in mRNA expression of the target gene affects the survival rate, molting, and wing expansion, healthy 1-day-old fourth-instar nymphs of *S. furcifera* were selected for RNA silencing, and each nymph was injected with 0.1 μL (approximately 100 ng) of dsRNA or water. The RT-qPCR results revealed that, compared with the ds*GFP*-injected insects, the expression of *SfDicer2* in the ds*SfDicer2* treatment group decreased by 90.50% after 24 h and by 82.14% after 48 h; the interference efficiency after 72 h was 68.37% (Figure 3A, Appendix A). Compared with the expression in the two control (ds*GFP* or water) groups, the target gene expression in the experimental group (ds*SfDicer2*) was significantly downregulated (*p* < 0.05). After 8 days, the survival rate of the experimental group was significantly lower than that of the two control (ds*GFP* or water) groups (*p* < 0.001; Figure 3B, Appendix A). Compared with the results in the two control (ds*GFP* or water) groups (Figure 3C,F, Appendix A), the mortality rate in the ds*SfDicer2* treatment group was 42.69% (Figure 3D,E, Appendix A). In particular, 35.62% of the deaths were attributable to the failure of epidermal dehiscence and molting. Furthermore, the mortality rate associated with wing deformity was 7.30%. After *SfDicer2* RNAi, the wings of some *S. furcifera* insects curled or folded (Figure 3G,H); the wing deformity rate of the surviving insects was 37.08% (Table 2).

### 3.4. Effects of SfDicer2 Silencing on the Gene Expression of Chitinase, Chitin deacetylase, Trehalase, and Chitin synthase in S. furcifera

The samples were collected after silencing the *SfDicer2* via RNAi for 48 h. To clarify the mechanism underlying the effect of silencing *SfDicer2* on chitin synthesis and degradation in *S. furcifera*, we evaluated the expression of the genes associated with chitinase (*SfCht5*, *SfCht7*, *SfCht10*, and *SfIDGF2*), chitinase deacetylase (*SfCDA1*, *SfCDA2*, *SfCDA3*, and *SfCDA4*), trehalase (*SfTRE1* and S*fTRE2*), and chitin synthase 1 (*SfCHS1*, *SfCHS1a*, and *SfCHS1b*) in *S. furcifera* (Figure 4, Appendix A). The results indicated that after *SfDicer2* was silenced for 48 h, the expression of *SfCht5*, *SfCht7*, *SfCht10*, and *SfIDGF2* decreased significantly compared with that of the control (ds*GFP*) group (*p* < 0.05; Figure 4A). Similarly, the relative transcript levels of *SfCDA2*, *SfCDA3*, and *SfCDA4* were significantly lower than those in the control (ds*GFP*) group; however, the relative expression of *SfCDA1* did not vary significantly between the groups (Figure 4B). In addition, the relative expression of both trehalase (*SfTRE1* and *SfTRE2* (Figure 4C)) and three chitin synthase 1 genes (*SfCHS1*, *SfCHS1a*, and *SfCHS1b* (Figure 4D)) decreased significantly after *SfDicer2* silencing. These findings suggest that *SfDicer2* significantly affects the expression of chitinase, chitin deacetylase, trehalase, and chitin synthase 1 genes.

### 3.5. Analysis of the Effects of Silencing SfDicer2 on Wing Expansion-Related Genes

*SfDPP*, *SfHh*, *SfITP*, *SfNotch*, *SfSal*, *SfSPN5*, *SfWg*, and *SfAp* reportedly regulate wing expansion in insects. We used RT-qPCR to study the effect of silencing *SfDicer2* on wing expansion-related genes in *S. furcifera*. Compared with the findings in the control (ds*GFP*) group, the expression of *SfDPP*, *SfHh*, *SfITP*, *SfNotch*, *SfSal*, *SfSPN5*, *SfWg*, and *SfAp* decreased significantly 48 h after ds*SfDicer2* injection (*p* < 0.05 (Figure 5, Appendix A). The specific silencing of the target gene could affect the transcript levels of some wing expansion-related genes, indicating that the silencing of *SfDicer2* inhibits wing expansion.

## 4. Discussion

Dicer2 is a core element in the RNAi pathway and serves as a cleavage enzyme in the siRNA pathway. An amino acid sequence characterization and phylogenetic tree analysis in *S. furcifera* revealed that the Dicer2 protein of *S. furcifera* was most closely related to that of *N. lugens* and *L. striatellus*. Previous studies have elucidated that Dicers of various species have similar domains, most of which include a HELICASE_ATP_BIND_1 or N-terminal DEAD domain, a HELICASE_CTER domain, a DICER_DSRBF domain, a PAZ domain, two RNase III domains, and a dsRBD domain [25,26]. The activity of helicases is crucial for processing dsRNA in the RNAi pathway [14]. The PAZ domain and tandem RNase III domains are responsible for excising siRNAs, preferentially from the ends of dsRNA molecules [27]. dsRBD is a typical domain of Dicer found in *D. melanogaster* [28]. Our results demonstrated that *S. furcifera* Dicer2 contains seven conserved domains similar to those in *N. lugens* and *L. striatellus*. This indicated that the protein is highly conserved in planthoppers.

Dicer2 is an endoribonuclease responsible for fragmenting exogenous dsRNA into siRNA, and it plays an essential role in the RNAi antiviral immune mechanism [26,29]. In this study, the expression of *SfDicer2* in various developmental stages and tissues of *S. furcifera* was detected using RT-qPCR. The results showed that the target gene can play an important role in the growth and development of *S. furcifera*. However, the observed expression patterns were inconsistent with those in other insects. For instance, a study on *D. melanogaster* revealed that *Dicer2* was expressed in all developmental stages and that its transcript levels were higher in pupae and adults in the late developmental stage than those in the early developmental stage. However, the expression of *Dicer2* was lower in the nymphal stage than in the adult stage, which may be attributable to the different expression patterns of *Dicer2* in different insects [16]. The evidence suggests that the third instar is a sensitive period of wing morph determination in *N. lugens*, and the wing bud development of the third-instar nymph determines the length of the fifth-instar wing bud [30]. The period between 24 and 36 h of molting in the fifth instar is a sensitive period for wing morph determination in *N. lugens* males and females, respectively [31]. This is consistent with the results of our study. Along with its high expression in 1-day-old first-instar nymphs and 2- and 3-day-old third-instar nymphs, *SfDicer2* exhibited high transcript levels in 1- and 3-day-old fifth-instar nymphs, suggesting that *SfDicer2* can play a vital role in wing differentiation. Furthermore, *SfDicer2* participates in the growth, development, and reproduction of insects. The relative transcript level of *SfDicer2* was high in the fat body, head, and integument, and the gene was distributed in the other tissues of *S. furcifera*. The fat body participates in the growth, development, and reproduction of insects as a crucial energy store and as a metabolic organ [31]. Therefore, this gene may participate in the growth and development of *S. furcifera*.

dsRNA is fragmented by *Dicer2* to generate a 12–23-bp siRNA [32,33]. The resulting siRNA combines with AGO2 to form RISC, thus prompting the silencing of the target gene [8,9]. Therefore, *Dicer2* plays a two-pronged role in the RNAi process. First, it functions upstream of RNAi to convert dsRNA to siRNA. Second, it functions downstream of RNAi for the formation of stable protein complexes (including Dicer2 and R2D2) after siRNA production, followed by mRNA shear [12,34]. The impact of dsRNA on the physiology of insects has also been studied in several species. It was found that dsRNA triggers the upregulation of RNAi and other immune-related genes in insects. Upon the delivery of (exogenous) dsRNA, the gene expression of *Dicer2* is particularly upregulated to 395 times in *Manduca sexta* [35], 5 times in *Blattella germanica* [36], and 2.3 times in *Acyrthosiphon pisum* [37]. Therefore, dsRNA treatment can affect *Dicer2* expression. To demonstrate whether dsRNA affects *Dicer2* expression, we set up ds*GFP* and water as controls. Obviously, the relative mRNA expression level of *SfDicer2* in both the ds*GFP* treatment group and the nuclease-free water treatment group were basically consistent with no significant indigenous change, indicating that treatment with dsRNA had no effect on the expression of *Dicer2*.

As a key effector of the RNAi-silencing mechanism, *Dicer2* has considerable significance in the life activities of insects [15,16]. Previous studies have detected *Ago2* and *Dicer2* transcripts in the eggs, first-/second-instar larvae, third-/fourth-instar larvae, pupae, and male and female adults of *Aedes albopictus*, suggesting that siRNA-mediated RNAi plays a regulatory role in its growth and development [38]. In this study, *SfDicer2* RNAi was associated with significant mortality and deformity, contradicting the previous findings that *Dicer2* had no specific effect on the growth and development of *N. lugens* [17]. In a study on *N. lugens*, RT-qPCR results revealed that the transcript level of *NlDicer2* was lowest (55%) on the first day after feeding the third-instar nymphs for 4 days. Only 7% of the insects died after being fed ds*Dicer2* for 4 days, and an 80% survival rate was achieved by *N. lugens* after 8 days without any notable deformity. These results could be explained by the high interference efficiency achieved with the injection method in this study. At 24 and 48 h after the injection of ds*PxDicer2* into *Plutella xylostella*, the expression of the target genes decreased by 73.1% and 73.6%, respectively, compared with the expression at 0 h. Moreover, the mortality rate reached 77%. However, no obvious phenotypic changes were observed [39]. Some studies also revealed that silencing *Drosha*, *Dicer2*, *Pasha*, and *Ago1* is not conducive to the transformation of *Diabrotica virgifera virgifera* larvae to adults [40]. The expression of these four transcripts increased from the third-instar stage to the pupal stage in this insect. In most cases, treatment with dsRNA before this stage reduced the protein’s abundance to below the level required to complete metamorphosis [41]. According to the binding efficiency of *Dicer2* to dsRNA or the cleavage efficiency of mRNA in different insects, the degradation efficiency of mRNA varies [42], resulting in different outcomes.

Chitin is an important component of an insect’s exoskeleton, and its synthesis and degradation are crucial for insects to successfully molt. Studies revealed that silencing chitin synthase 1 is fatal to *Locusta migratoria*, which is attributable to molting difficulties [43]. Tang et al. studied the function of the trehalase gene in *Tribolium castaneum* and silenced it using RNAi technology, which resulted in molting deformity and death [44]. *GpTre*1 and *GpTre2* play significant roles in the growth of *Glyphodes pyloalis* by influencing chitin metabolism [45]. Similarly, silencing the trehalase genes (*TRE1* and *TRE2*) inhibits the growth of *N. lugens* [46]. Deacetylases play a vital role in insect molting, and interfering with their expression may lead to molting disorders and death [47]. For instance, *SfCDA1*, *SfCDA2*, and *SfCDA4* play crucial roles in the molting of *S. furcifera* [48]. *SfCht5*, *SfCht7*, *SfCht10*, and *SfIDGF2* also play essential roles in the molting of *S. furcifera*. Silencing the *SfCht5*, *SfCht7*, *SfCht10*, and *SfIDGF2* genes resulted in the downregulation of chitin synthesis and decreased expression of chitin deacetylase genes, which subsequently affected the normal growth and development of insects [49]. These data are in agreement with the observed mortality rate of *S. furcifera* after the silencing of *SfDicer2* in this study, including the high number of deaths in which the epidermis did not split or the insect became slender and molting failed. The transcript levels of four chitinase, two chitin deacetylase, two trehalase, and three chitin synthase 1 genes in *S. furcifera* decreased significantly after *SfDicer2* was silenced. Therefore, we speculated that *SfDicer2* affects the molting of *S. furcifera* by regulating the expression of chitinase, chitin deacetylase, trehalase, and chitin synthase 1 genes.

Insect wing expansion is also a crucial biological process regulated by multiple genes and signaling pathways. The development and differentiation of *D. melanogaster* wing primordium are mainly regulated by the Hedgehog (*Hh*), Decapentaplegic (*Dpp*), and Wingless (*Wg*) genes [50,51]. Morgan et al. first identified the gene *Notch* in *D. melanogaster*. Wing defects were observed when the partial function of *Notch* was missing [52]. After ds*Notch* injection, 10% of *L. migratoria* specimens displayed wing disorders after molting [53]. The elongation of insect wings is regulated by other genes, such as ion transport protein (*ITP*) and serpin-5 (*SPN5*) [54,55]. For example, *ITP* regulates wing expansion in *N. lugens* [56]. After *Dpp* expression, the wings of *N. lugens* exhibited different degrees of wing deformity [57]. In a study on *S. furcifera*, the silencing of *Dpp* resulted in abnormal phenotypes, such as curling or folding [58]. Similarly, *Wg* may affect wing elongation in *S. furcifera*. After the administration of ds*Wg* in *S. furcifera*, 65.63% of the wings curled or failed to stretch normally [59]. Similar observations were observed in our study, which suggests that *SfDicer2* plays an important role in regulating wing expansion in *S. furcifera*. Seven wing expansion-related genes (*SfDPP*, *SfHh*, *SfITP*, *SfNotch*, *SfSal*, *SfSPN5*, *SfWg,* and *SfAp*) were significantly downregulated after *SfDicer2* was silenced. Therefore, *SfDicer2* may affect wing expansion by regulating the expression of wing expansion-related genes.

These results provide a theoretical basis for the use of *Dicer2* as a target gene for pest control in future studies.

## 5. Conclusions

We identified *SfDicer2* using the genomic and transcriptomic data of *S. furcifera*. An RT-qPCR revealed that *SfDicer2* is expressed in different developmental stages and tissues of *S. furcifera*. Moreover, this gene may be associated with the growth and development of the insect. The silencing of *SfDicer2* resulted in significant inhibition of molting and wing expansion in *S. furcifera* and the downregulation of chitinase, chitin deacetylase, trehalase, and chitin synthase 1 and wing expansion-related genes. The results of our research may help clarify the spatiotemporal expression characteristics and biological functions of *SfDicer2* to further understand its internal molecular mechanism in the regulation of insect growth and development. They can also provide a theoretical basis for future studies on the use of *Dicer2* as a target gene for pest control.

## Figures and Tables

**Figure 1 insects-13-00677-f001:**
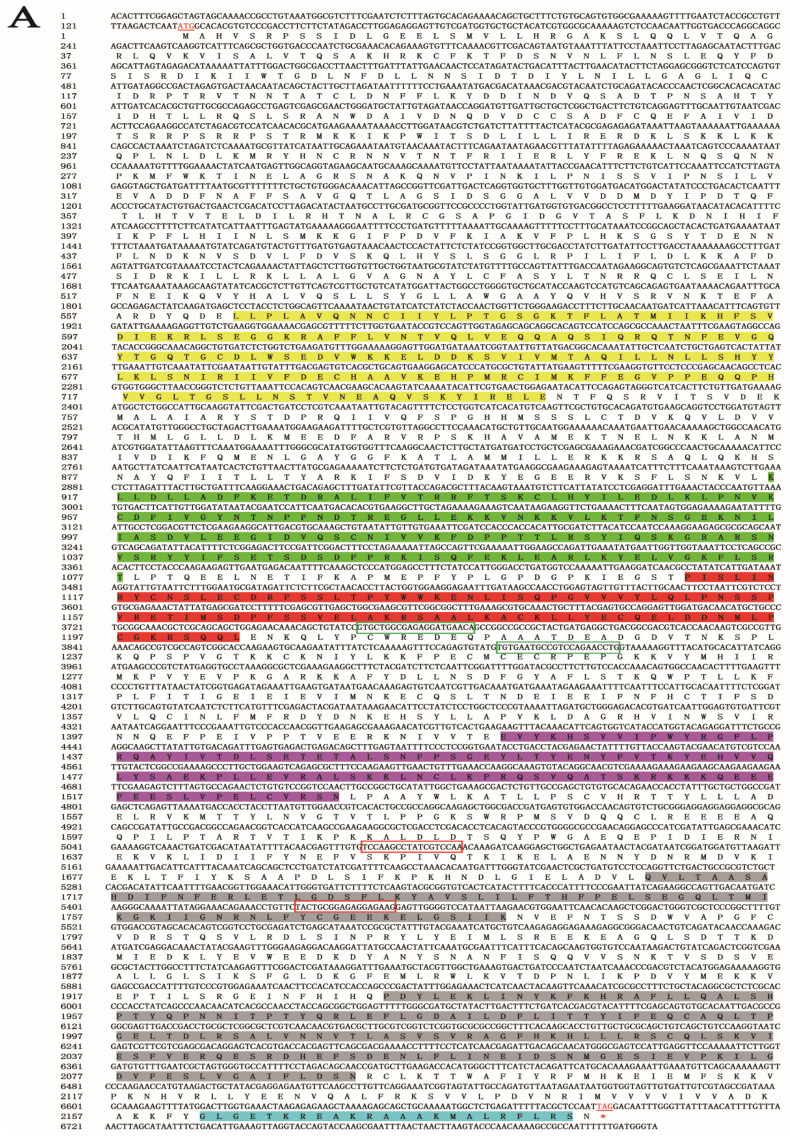
Bioinformatic analysis of *SfDicer2* in *S. furcifera*. (**A**) Analysis of nucleotide sequence of *SfDicer2* and amino acid sequence of *SfDicer2*. Underlined and red font ATG, start codon; underlined and red font TGA with an asterisk, stop codon; yellow background, HELICASE_ATP_BIND_1; green background, HELICASE_CTER; red background, DICER_DSRBF; purple background, PAZ; gray background, RNase III; and blue background, dsRBD. The green boxes represent the primers for RT-qPCR, and the red boxes represent the primers for dsRNA synthesis. (**B**) Phylogenetic analysis of *SfDicer2* homologs obtained from insect species based on amino acid sequences. The sequences were downloaded from the GenBank protein database. The red star indicates the Dicer2 protein of *S. furcifera*. (**C**) Conserved domain analysis of Dicer2 proteins. The different shapes and colors represent different protein domains (green pentagon, HELICASE_ATP_BIND_1 domain; orange hexagon, HELICASE_CTER; dark blue rectangle, DICER_DSRBF domain; orange pentagon, PAZ domain; dark blue oval, RNase III domain; and green hexagon, dsRBD). A conserved domain analysis was performed using the ScanProsite online server.

**Figure 2 insects-13-00677-f002:**
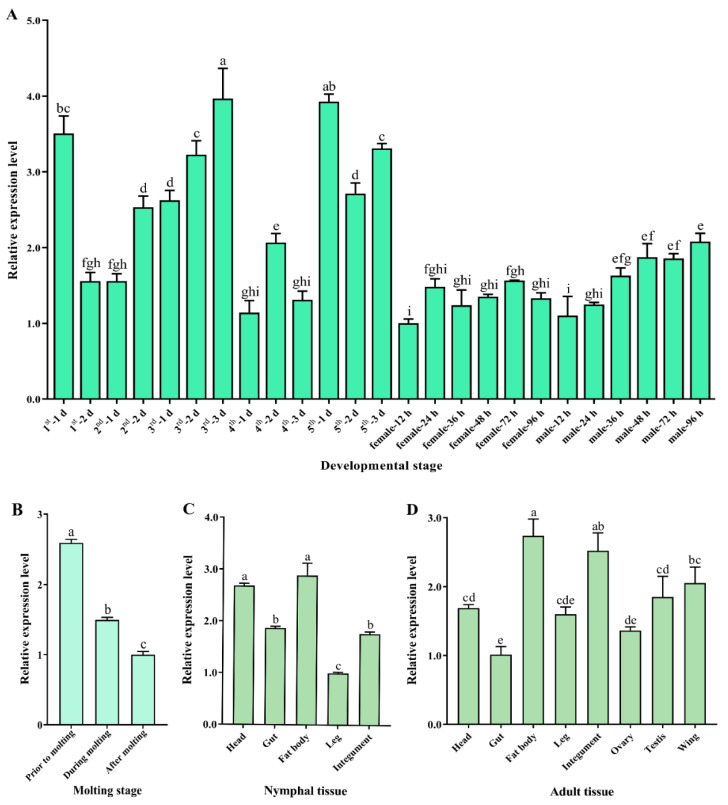
Relative expression of *SfDicer2* in various developmental stages and tissues of *S. furcifera*. *SfRPL9 and SfTUB* were used as internal control genes. (**A**) Relative expression of *SfDicer2* from day 1 in first-instar insects to adulthood in males and females at 96 h after eclosion, as determined using RT-qPCR. (**B**) Relative expression of *SfDicer2* before, during, and after molting, as determined using RT-qPCR. (**C**) Relative expression of *SfDicer2* in different tissues of the fifth-instar nymphs, as determined using RT-qPCR. (**D**) Relative expression of *SfDicer2* in different tissues of 1-day-old male and female adults, as determined using RT-qPCR. The expression levels of *SfDicer2* in 12-hours-old female adults, after molting, in nymphal legs and adult gut were used as the references. Data in the figure are presented as the means ± SEs of three replicates. Different letters above the bars indicate significant differences in gene expression among the different developmental stages or different tissues (Duncan’s multiple range test, *p* < 0.05).

**Figure 3 insects-13-00677-f003:**
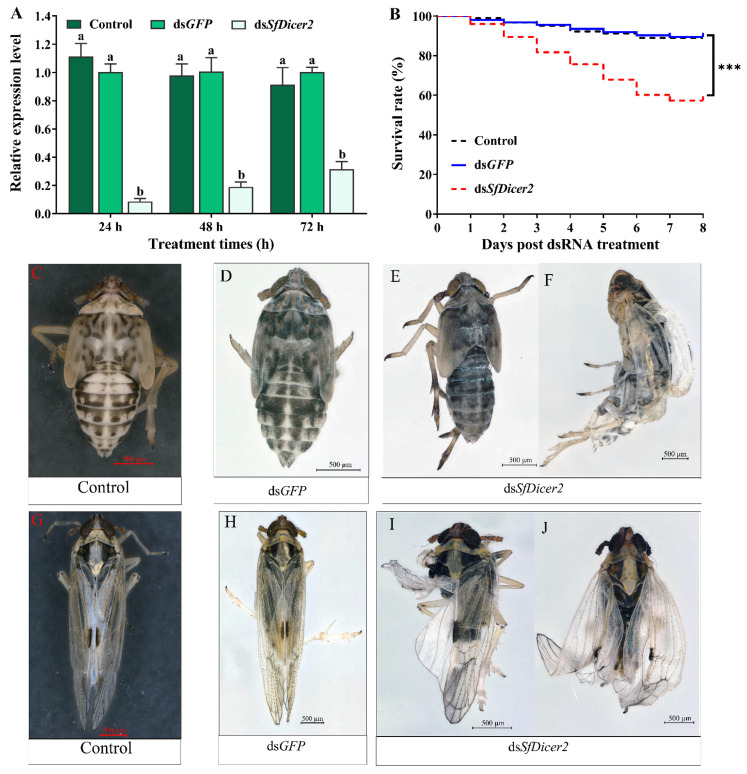
Comparison between the *S. furcifera* treatment (ds*SfDicer2*) and the two control (ds*GFP or water*) groups. (**A**) Effects of *SfDicer2* silencing at the transcriptional level. Data in the figure are presented as the means ± *SE*s of three replicates. The significance of the differences between the treatment (ds*SfDicer2*) and control (ds*GFP* or water) groups was determined using Student’s *t*-test for independent samples. (**B**) Survival rates after eight consecutive days. (**C**,**D**) Normal phenotypes of 3-day-old fifth-instar nymphs of *S. furcifera* (before molting) in the two control (ds*GFP* or water) groups. (**E**) Abnormal phenotype (lengthening body, ventral retraction, and inability to molt) of 3-day-old fifth-instar nymphs of *S. furcifera* (before molting) in the treatment group (dsSf*Dicer2*). (**F**) *S. furcifera* failed to fully molt and died in the treatment group (ds*SfDicer2*). (**G**,**H**) Normal wings of adults in the two control (ds*GFP* or water) groups. (**I**,**J**) Curled or folded wings in the treatment group (ds*SfDicer2*). Different letters above the bars indicate significant differences in gene expression among the three groups of experimental conditions (Duncan’s multiple range test, *p* < 0.05); Significant differences between the treatment and control (ds*GFP*) groups are indicated using asterisks (***, *p* < 0.001).

**Figure 4 insects-13-00677-f004:**
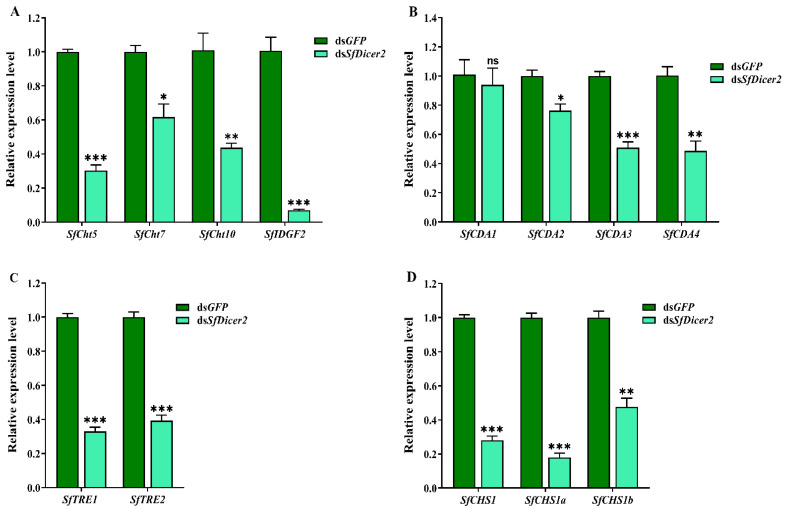
Effects of silencing *SfDicer2* on the expression of chitinase, chitin deacetylase, trehalase, and chitin synthase 1 genes. (**A**) The expression levels of chitinase genes: ds*SfCht5*, ds*SfCht7*, ds*SfCht10*, and ds*SfIDGF2*. (**B**) The expression levels of four chitin deacetylase genes: *SfCDA1*, *SfCDA2*, *SfCDA3*, and *SfCDA4*. (**C**) The expression levels of trehalase: *SfTRE1* and *SfTRE2*. (**D**) The expression levels of chitin synthase 1 genes: *SfCHS1*, *SfCHS1a*, and *SfCHS1b*. Data in the figure are presented as the means ± SEs of three replicates. The significance of the differences between the treatment (*dsSfDicer2*) and control (ds*GFP*) groups was determined using a Student’s *t*-test for independent samples. The asterisks above the bars indicate significant differences (*, *p* < 0.05; **, *p* < 0.01; ***, *p* < 0.001; ns, no significance difference).

**Figure 5 insects-13-00677-f005:**
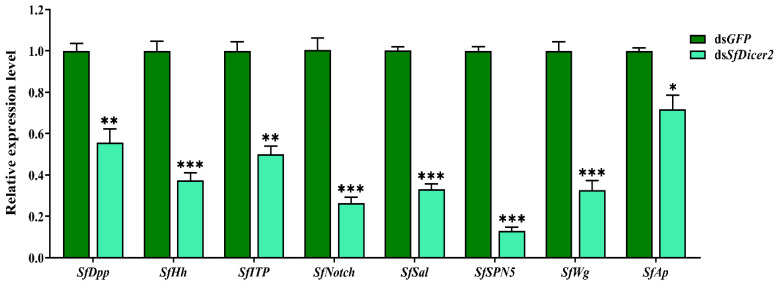
Effects of silencing *SfDicer2* on the expression of wing expansion-related genes. The data in the figure are presented as the means ± SEs of three replicates. The significance of the differences between the treatment (*dsSfDicer2*) and control (ds*GFP*) groups was determined using a Student’s *t*-test for independent samples. The asterisks above the bars indicate significant differences (*, *p* < 0.05, **, *p* < 0.01; ***, *p* < 0.001).

**Table 1 insects-13-00677-t001:** Primers used for a real-time quantitative polymerase chain reaction (RT-qPCR) and dsRNA synthesis.

Purpose	Primer Name	Primer Sequence (5′–3′)
RT-qPCR analysis	*SfDicer2*-F	GTGCTGGCGAGAGGATGAACA
	*SfDicer2*-R	CAGGTTCTGGACGGCATTCACA
	*SfCht5*-F	GAACTACGCACAGCCTTCCTCAA
	*SfCht5*-R	CCAGTTGCCTCTCAAGTCATACGA
	*SfCht7*-F	CGGAATGCCAACTTACGGACGC
	*SfCht7*-R	CGCAGCATCTCGCACACCTCATAG
	*SfCht10*-F	GCCGAATACCTGGACTGGATCTCT
	*SfCht10*-R	TCAAGCCGTGCTGCTCCTTCT
	*SfIDGF2*-F *SfIDGF2*-R	TGTCAACCGACGCACCTTCCT CGAGACCCTGATACGCTGAGAGT
	*SfCDA1*-F	TCACATTCAACGGTGCCATCAAC
	*SfCDA1*-F	GCCATCTCAGCCAACCAATCATC
	*SfCDA2*-F	TCCATCACGCACAATGACGAAGAA
	*SfCDA2*-R	GAACTGGTTGTTGCCTCCGACTC
	*SfCDA3*-F	ACACAGCCGCATGTCAACTACC
	*SfCDA3*-R	ATCTCGCAGCCATTCGGATTACG
	*SfCDA4*-F	TCCGATTGCCGCCACCTTCTAT
	*SfCDA4*-R	CTCAAGACGCACTCCACCATAAGC
	*SfTRE1*-F	GACTTCTGCTATGTGATATGC
	*SfTRE1*-R	GCTGTCCACCATCTGAATA
	*SfTRE2*-F	GTGGTTGGATGCTGTTACTA
	*SfTRE2*-R	GAGATGTTTGTCGGGTAGAA
	*SfCHS1*-F	GATTGGTCATTGGCTTCAGA
	*SfCHS1*-R	GTAATGTCTTGCTTCGTCAG
	*SfCHS1a*-F	CTTCGGTGTTTGGTTTCTT
	*SfCHS1a*-R	TGGGTAACATCATCATAGGA
	*SfCHS1b*-F	GAGAAGGCGAGAATAGCA
	*SfCHS1b*-R	GCAGCAAGAACACGATTA
	*SfDpp*-F	CGCAGTCGTGTCGTTGTTCCT
	*SfDpp*-R	TCTGGTCGATGTCGCTCTCCTTAT
	*SfHh*-F	GCGACAGCCAGGTGACAACAT
	*SfHh*-R	CGTAGAAGGAGATGCGAGGGTGT
	*SfITP*-F	TTCAGTGCAAGGGTGTCTACG
	*SfITP*-R	AGCGAATGTAGTTGAGCCTCT
	*SfNotch*-F	CGCCGCCTAGTATGGAGACAGT
	*SfNotch*-R	CTATGCCGTCGTCAATGGAAGGATC
	*SfSPN5*-F	AAGTGGAACAGACAAAGACAGAGGAA
	*SfSPN5*-R	TCATCACTCTTCTTCTTCTCATCCT
	*SfSal*-F	TCTTCGCAGTCTTCCAGTATCACAC
	*SfSal*-R	CTCTGAGCCACTTGCCACTGTC
	*SfWg*-F	CAAGAAGAACCGCTACAACT
	*SfWg*-R	GATGACTTCACAGCACCAG
	*SfAp*-F	GGAAACGCAAGCCCAAGGAT
	*SfAp*-R	TCATCATCATTCGCCGCCATT
	*SfRPL9*-F	GGGCGAGAAGTACATCCGTAGG
	*SfRPL9*-R	GCGGCTGATCGTGAGACATCTT
	*Sf**TUB*-F	CGCTGTTGATGGAGAGGCTGTC
	*Sf**TUB*-R	ACGACGGCTGTGGATACCTGTG
dsRNA synthesis	*Dicer2*-F	TAATACGACTCACTATAGGGTCCAAGCCTATCGTCCAA
	*Dicer2*-R	TAATACGACTCACTATAGGGCTTCTCCTCTCCGCAGTA
	*GFP*-F	TAATACGACTCACTATAGGGGCCAACACTTGTCACTACTT
	*GFP*-R	TAATACGACTCACTATAGGGGGAGTATTTTGTTGATAATGGTCTG

Note: Underlined nucleotides indicate the T7 promoter.

**Table 2 insects-13-00677-t002:** Effects of *SfDicer2* RNAi on the molting and wing expansion of *S. furcifera.*

Gene	Death		Survival	
	Mortality (%)	Lethal Phenotype Rate (%)	Survival Rate (%)	Wing Deformity Rate (%)
Control	11.11 ± 0.56 a	00.00 ± 0.00 a	88.89 ± 0.56	1.25 ± 0.62 a
ds*GFP*	10.56 ± 0.37 a	00.00 ± 0.00 a	89.44 ± 0.37 a	1.37 ± 0.46 a
ds*SfDicer2*	42.69 ± 0.91 b	42.92 ± 1.68 b	57.31 ± 0.91 b	37.08 ± 0.63 b

Note: Data are presented as the means ± SEs of three replicates. Different letters indicate significant differences among the three groups of experimental conditions (Duncan’s multiple range test, *p* < 0.05).

## Data Availability

All data used/generated in this study have been included in the text and Appendix A of this manuscript.

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
