# Peer review of "SfDicer2 RNA Interference Inhibits Molting and Wing Expansion in Sogatella furcifera"

_insects, 2022, doi:10.3390/insects13080677_

Round 1

Reviewer 1 Report

Comments:

Endoribonuclease 2 (Dicer2) is a key nicking endonuclease involved the small interfering RNA biosynthesis pathway, and it plays important roles in gene regulation and viral immunity. However, the role of SfDicer2 in the regulation of molting and wing expansion in S. furcifera is unknown. Here, Zeng et al., has found that SfDicer2-mediated RNAi result in wing deformities and lethal modifications in S. furcifera through the significant inhibition of the chitin synthesis and degradation pathways and wing expansion genes. The study is of interest and important. The manuscript is proper to be published in insects with careful revision. 

Other points that may be revised:
1) English writing should be further edited. 

2) In Figure 1, Y axis in A and B has two numbers, but Y axis in C and D has one number. The author should unify the formation. 

3) In results 3.2, why the author chooses 1-day-old fourth-instar nymphs of S. furcifera to perform SfDicer2 RNA silencing experiment. According to the Figure 1A, the relative SfDicer2 expression is lowest at this stage. It may mean that the function of SfDicer2 in this stage is relative weak. So, the reviewer wants to know the author’s consideration. 

4) In Figure 2B, “ds GFP vs. ds SfDicer2 :P<0.001”, this legend is not standard. Because it is not every timepoint P<0.001. The author need label the related “*”, “**” or “***” in the related timepoint. 

5) ds GFP also need to use “Italics” in the whole manuscript. 

6) In the main text, the author dose not quote Figure 2F. 

7) The Figure 2C and 2F, they both inject same ds GFP to S. furcifera, but they have big difference from the appearance. It is caused by different sex or stage, or other reseasons? 

8) The reviewer does not understand the table in line 230. The reviewer guess it means the original data after performing the ds GFP and ds SfDicer2 injection. And this data can remove to the supplemental materials. 

9) In results 3.4, the title font is not bold. This sentence “To clarify the mechanism underlying the effect of SfDicer2 silencing on the reproduction of S. furcifera” in line 242, the “reproduction” may be replaced with “chitin synthesis and degradation”. The sentence “chitinase deacetylase 1 (SfCDA1, SfCDA2, SfCDA3, and SfCDA4)” in line 245 and “three chitin deacetylase 1 genes (SfCHS1, SfCHS1a, and SfCHS1b) also significantly decreased after SfDicer2 silencing” in line 252, the same chitin deacetylase 1 contains different genes. The author must check which one is correct. In the sentence “chitin synthesis pathways (SfTRE1, SfTRE2, SfCHS1, SfCHS1a, and SfCHS1b) in S. furcifera” in line 246., why two trehalose genes (SfTRE1 and SfTRE2) belong to chitin synthesis pathways. 

10) Because that Dicer2 is involved the small interfering RNA biosynthesis pathway that mediates gene silencing. But the authors focus on the genes downregulated by silencing Dicer2, not on upregulated genes. Why.

Author Response

Dear reviewer,

         Thank you very much for yourYour suggestions of our article. Please see the attachment

Reviewer 2 Report

SfDicer2 RNA Interference Inhibits Molting and Wing 2 Expansion in Sogatella furcifera manuscript is very interesting. I enjoyed very much reading it. The study is relevant to the field, especially for hemipteran insects, and the manuscript, on average, is well written. 
I have some suggestions to improve the manuscript. 

Format:

-Check the gene abbreviation. It must be italicized;

- Panel B, C and D from Fig. 2 are out of alignment.

- Charts from Fig. 3 are out of alignment.

- Check Tables captions; Table 2 does not have a caption, and table 3 is labeled as 2.

-Lagend for figure 2 is poorly described. There is missing information about the gene expression experiments, survival ship analysis and mutant phenotypes

- Lagend for figures 3 and 4 are poorly described. There is missing information about the gene expression, such as the age of insects and biological repetitions

- Reference 15 on line 284 should be on line 283 after the sentence “the pupae and adults in the late development stage.”.

Methods

-What is the initial concentration of RNA used for the cDNA synthesis?

-Is the cDNA diluted prior to the RT-qPCR analysis?

-What is the source of the eGFP gene? Plasmid?

Results & discussion

-The authors informed that DICER -2 gene were retrieved from transcriptomic and genomic data; however, none of the cited papers describes the SfDicer2. Since this manuscript aims to highlight some functions of this gene, it would be interesting to describe the main feature of this gene, such as structure, and phylogenetic comparisons between SfDicer2 and other insects.

-Since the third and fifth instars showed the highest expression of SfDicer2, why the fourth instar was chosen for the RNAi experiments?

- What is the lethal phenotype reported in table 3?

Author Response

Dear reviewer,

         Thank you very much for your suggestions of our article. Please see the attachment

Reviewer 3 Report

Major points:

Firstly, I would like to recognize the great amount of work done by the authors. The samples obtained can certainly lead to valuable data, if analysed correctly. There are however some experimental design flaws that greatly compromise the validity and/or significance of the obtained results.

 1) Lack of proper reference genes.

The bulk of results presented in this manuscript are relative transcript levels assayed via RT-qPCR. While a powerful technique, it requires careful selection of reference genes and quality control to ensure that the expression levels of the reference genes are stable throughout the experimental conditions. The authors opted to use a single (RPL9) gene as an internal reference. They support their decision with a citation (An, Hou and Liu 2016). The paper they cite however demonstrates that in Sogatella furcifera: (1) no single gene or combination of genes tested can be used for both inter-developmental stage and inter-tissue analyses; (2) that more than one reference gene should be used to reduce biases during normalization; (3) that to avoid biased results multiple analytical tools should be employed to assay the reference gene stability.

Furthermore, no data or citations were presented that elucidate how RPL9 is affected by the experimental conditions applied in the manuscript (injection of dsRNA).

The results presented in this manuscript can therefore not be considered reliable. The authors are advised to test the stability of, and then make use of, multiple reference genes as described in An, Hou and Liu 2016.

2) Inappropriate control conditions

The authors seek to investigate the impact of a Dicer2 knockdown on various physiological and developmental processes by delivering a dsRNA construct complementary to Dicer2 via injection. As a negative control, they inject a dsRNA construct complementary to GFP. Next, the authors assay the degree of Dicer2 knockdown via RT-qPCR. This is a flawed experimental setup, as Dicer2 is known to be upregulated by the delivery of dsRNA. Injecting dsGFP might therefore increase the Dicer2 transcript levels compared to an untreated animal. Consequently, upon injection of dsSfDicer2, a RT-qPCR using dsGFP treated animals for normalization cannot determine if the Dicer2 transcript levels are up or downregulated compared to an untreated animal. Thus, despite significant differences in transcript levels between dsGFP and dsSfDicer2 treated specimens, no statement can be made regarding an up- or downregulation of Dicer2 following treatment with dsSfDicer2. Consequently, the phenotypical effects cannot be associated with Dicer2 with certainty.

To overcome this issue, the authors are advised to: (1) determine whether/to which degree Dicer2 is affected by the delivery via injection of dsRNA; (2) make use of a negative control without dsRNA (e.g. saline injection); (3) determine actual knockdown levels at the protein level (e.g. via a western blot).

3) No consideration for off-target effects

As mentioned in my previous point, the authors attempt to induce a knockdown of Dicer2 by injecting a dsRNA construct complementary to this gene. They do not describe what measures were taken to verify the specificity of the designed construct to avoid off-target effects. As described by Chen et al. 2021 (second reference in the list of this manuscript), long dsRNA can lead to off-target effects which need to be accounted for. Consequently, it is not possible to determine with certainly a causal link between a knockdown of Dicer2 and the observed effects.

This issue can be addressed by carrying out the experiments with two non-overlapping dsRNA constructs in parallel. If the same experimental results are obtained when using both constructs a strong case can be made that they are Dicer2 dependent.

Minor points:

1) The figure and table captions need to be revised. The caption of figure 2 is misplaced and the caption of table 1 is missing.

2) The sequence of SfDicer2 identified by the authors should be provided. Furthermore, the identity of the sequence should be confirmed through a phylogenetic analysis.

3) The authors might want to discuss/speculate how knocking down Dicer2 leads to reduced transcript levels of other genes.  

4) In table 2 the mortality rate and survival rate are both provided. This seems superfluous as they describe the same parameter and can simply be derived from each other.

5) Additional data, such as details of the statistical analysis results, GFP construct sequence, RT-qPCR raw Ct values etc. should be provided in the supplementary data.

Beside these three major points and minor points. I have made additional minor remarks in the uploaded PDF file. These should be addressed as well.

Author Response

(The authors gave the same response as above.)

Round 2

Reviewer 1 Report

The authors have addressed my questions.

Author Response

Thank you for your previous questions, other questions have been revised in the revised manuscript.

Reviewer 3 Report

The reviewer has considered the responses given by the authors to the major points brought forward in the first review report.

An analysis of the strenghts and flaws of the responses; reviewer suggestions; and minor comments on the second manuscript draft, are given in the attached document.

Author Response

Thanks for your suggestion, we have supplemented experimental data or answered your question. Please see the attachment.

Round 3

Reviewer 3 Report

See the attached PDF for the major and minor remarks.

Author Response

Dear reviewer,

         First of all, we sincerely appreciate your valuable comments on our experimental design and article. We have carefully revised the problems existing in our article, and the specific revisions can be found in the manuscript and attachment.
